# Epidemiological/Disease and Economic Burdens of Cervical Cancer in 2010–2014: Are Younger Women at Risk?

**DOI:** 10.3390/healthcare11010144

**Published:** 2023-01-03

**Authors:** Chuhao Xi, Jay J. Shen, Betty Burston, Soumya Upadhyay, Shoujun Zhou

**Affiliations:** 1Jiangsu Cancer Hospital & Jiangsu Institute of Cancer Research & The Affiliated Cancer Hospital of Nanjing Medical University, Nanjing 210009, China; 2Department of Healthcare Administration and Policy School of Public Health, University of Nevada, Las Vegas, NV 89154, USA; 3School of Health Policy & Management, Nanjing Medical University, Nanjing 211100, China

**Keywords:** cervical cancer, epidemiological burden, economic burden

## Abstract

Objective: Cervical cancer is an important factor threatening women’s health in China. This study examined the epidemiological and economic burden of cervical cancer among the medically insured population, which could provide data support for government departments to formulate policies. Methods: All new cases of cervical cancer under the Urban Employee Basic Medical Insurance (UEBMI) plan in a provincial capital city in eastern China from 2010 to 2014 were collected. The Cox proportional hazard model was used to analyze the factors affecting the survival rates for cervical cancer. Outpatient and hospitalization expenses were used to assess the direct economic burden, and the Potential Years of Life Loss (PYLL) and potential economic loss were calculated by the direct method to assess indirect burden. Results: During the observation period, there were 1115 new cases and 137 deaths. The incidence rate was 14.85/100,000 person years, the mortality was 1.82/100,000 person years, and the five-year survival rate was 75.3%. The age of onset was mainly concentrated in the 30–59 age group (82.9%) and the tendency was towards younger populations. The age of onset (HR = 1.037, 95% CI = 1.024–1.051), the frequency of hospitalization services (HR = 1.085, 95% CI = 1.061–1.109), and the average length of stay (ALOS) (HR = 1.020, 95% CI = 1.005–1.051) were the related factors affecting overall survival. Among the direct economic burden, the average outpatient cost was $4314, and the average hospitalization cost was $12,007. The average outpatient and hospitalization costs within 12 months after onset were $2871 and $8963, respectively. As for indirect burden, the average Potential Years of Life Loss (PYLL) was 27.95 years, and the average potential economic loss was $95,200. Conclusions: The epidemiological and economic burden reported in the study was at a high level, and the onset age of cervical patients gradually became younger. The age of onset, the frequency of hospitalization services and the ALOS of cervical cancer patients should be given greater attention. Policymakers and researchers should focus on the trend of younger onset age of cervical cancer and the survival situation within 12 months after onset. Early intervention for cervical cancer patients, particularly younger women, may help reduce the burden of cervical cancer.

## 1. Introduction

Cancer is one of the common causes leading to death in developing and developed countries, especially in China [1]. According to the statistics of “GLOBOCAN”, there were about 19,292,789 new cancer cases and 9,958,133 cancer deaths in the world in 2020 [2]. Among all cancers, cervical cancer, as an important factor threatening women’s reproductive health, has received increasing attention. Worldwide, cervical cancer ranks fourth for both incidence and mortality in carcinomas among women [2]; meanwhile, in China, cervical cancer ranks seventh for incidence and eighth for mortality in carcinomas among women [3]. In 2018, one third of the global disease burden of cervical cancer was contributed by China and India, with about 97,000 new cases and 48,000 deaths from cervical cancer in China [4]. The high rates of incidence and mortality in cervical cancer have brought a great burden to patients, their families and society. In the face of the rapidly increasing burden, the Chinese government has taken corresponding measures, especially in the diversification and efficiency of prevention and control. However, the rapid aging of the population and the increasing exposure to risk factors mean that prevention and control face greater challenges [5]. Although screening and early diagnosis for cervical cancer have improved over the past decade, it is still not common in China [6]. The economic burden of the follow-up treatment of cervical cancer and the corresponding losses cannot be ignored [7,8]. The research on the epidemiological and economic burden of cervical cancer is insufficient in China, which also means that the severity of cervical cancer and the changes in the cost structure after the onset need to be further elaborated.

During the past 20 years, several studies on trends of cervical cancer incidence have found younger women are easily affected by cervical cancer [9,10]. There is a geographical disparity in cervical cancer in the world, especially among Chinese cities, which leads to inadequate representativeness of research evidence. Meanwhile, there are a number of risk factors judged to be associated with cervical cancer prognosis, including stage at diagnosis, histology and age of onset [11,12]. Age of onset as an important prognostic factor for cervical cancer is a question that has been debated and evaluated in the literature without a clear, definitive answer [13,14]. Our study explored the trend of age of onset and analyzed the impact of age of onset and relevant influencing factors on the prognosis of cervical cancer patients to supplement the lack of evidence in this area.

Cancer classification research from specific populations could provide more clear and effective guidance for future policy [15,16]. Therefore, based on the population of the Urban Employee Basic Medical Insurance (UEBMI) plan in a provincial capital in eastern China, the aim of this study is to examine the epidemiological and economic burden of cervical cancer, to describe the trend of age of onset and to evaluate risk factors affecting the prognosis of cervical cancer. Findings from this study could provide key evidence for government departments to formulate improved policies.

## 2. Methods

### 2.1. Study Design and Subject Selection

According to Chinese law, employers and employees are obliged to pay social and health insurance. By 2011, the overall coverage of UEBMI in China reached more than 95% [17]. The medical and expense data of all insured personnel must be uploaded to the medical insurance management system, so as to ensure that the data is complete, accurate and representative. Cluster sampling was adopted in the study. All new cases of cervical cancer among the population of UEBMI in a provincial capital in eastern China from 1 January 2010 to 31 December 2014 were collected. All data of cervical cancer patients were uniformly extracted from the medical insurance system after the end of the observation period. Since the interval time was adequate and desensitization processing was carried out at the time of extraction, the data could be legally used and analyzed.

The selection criteria for new cases were based on the following principles. If there was a record of cervical cancer in the system, the case was directly included in the study. If there was no clear record in the system, the following three criteria were considered: 1. cervical cancer (according to ICD10) as discharge diagnosis; 2. special treatment records for cervical cancer in the system; 3. deaths due to cervical cancer. If one of the above three items was met, the case was included in the study.

### 2.2. Data Collection

Basic case information: Unique identification number of medical insurance personnel, date of birth, nationality, profession and total salary earned in the year of onset were collected from the system.

Disease-related information: The onset time of cervical cancer was determined according to the time when the onset time or the start time of special treatment for cervical cancer was recorded. The study determined whether there was metastasis according to the system registration records. In the observation period and within 12 months after the onset, the frequency of outpatient and hospitalization services and the average length of stay (ALOS) were also collected. In addition, the survival time of cervical cancer was calculated.

Cost information: In the observation period and within 12 months after the onset, respectively, outpatient costs, hospitalization costs and total medical costs (outpatient + hospitalization costs) were collected.

The observation period: the onset time of all new cases was between 1 January 2010 and 31 December 2014. However, because it was necessary to observe the survival status of the cases who were included the study in the final year, the observation time was ended on 25 September 2015, when a new medical service price reform was implemented in this city. The longest and the shortest observation periods were 5.73 years and 0.73 years, respectively.

Relevant population data: For data analysis, the number of enrollees in UEBMI in this city and the proportion of males and females were extracted.

### 2.3. Statistical Analysis

In the study, the mean and median were used to describe the central tendency. For non-normally distributed data, the Wilcoxon test was used for analysis. Incidence rate and mortality were used to describe epidemiological burden. Overall survival was estimated by Kaplan–Meier curves. The Cox proportional hazard model was used to analyze the factors affecting survival. All potentially relevant variables were first analyzed with a univariate Cox proportional hazard model to assess the associations. The criterion for selecting variables was set at a *p*-value less than 0.1 (90% level of significance) in the univariate analysis. Second, to control for potential confounders, a multivariate Cox proportional hazard model was performed. *p*-Values, hazard ratios (HR) and 95% CI were used to assess effect sizes. Economic burden was reported in 2014 US Dollars and 2014 Chinese Renminbi at a rate of 1 USD = 6.1238 RMB. For the present study, outpatient and hospitalization costs were used to assess the direct economic burden of cervical cancer. The potential economic loss was compared with the salary income in the year of onset. Potential Years of Life Loss (PYLL) and the potential economic loss that were calculated by the direct method were used to measure the indirect burden of premature mortality due to cervical cancer. All analyses were conducted using SPSS 22.0. The *p*-values of all the statistical analyses were two sided, and the level of significance for the multivariate analysis was 95% (*p* < 0.05).

## 3. Result

### 3.1. Demographic Characteristics

From 2010 to 2014, the total enrollees in UEBMI were 15.91 million person-years, of which women accounted for 7.51 million person-years. Table 1 shows the distribution of the demographic characteristics. During the observation period, there were 1115 new cases of cervical cancer, and the incidence rate was 14.85/100,000 person-years. A total of 137 persons died during the study, accounting for 12.3% of new cases, with a mortality rate of 1.82/100,000 person-years. The average age was 48.15 years, with a standard deviation of 11.01 years, and the median age was 47 years. Most of the cases of cervical cancer were in women aged between 30 and 59 years, with 36.4% at 30 to 44 years, and peaking at 46.5% for ages 45 to 59. As for ethnicity, 98.1% were of Han ethnicity. The highest salary income in the year of onset was $133,400, the lowest was only $637, and the median income was US $3323. Of the cases, 83.6% had an income of less than $5000. In addition, during the observation period, 56 cases had metastasis, accounting for 5% of the sample.

### 3.2. Utilization of Medical Services

Regarding outpatient services, 1043 (93.5%) cases received outpatient services. During the observation period, the average frequency of received outpatient services was 28.63. Within 12 months after the onset, 962 cases were treated as outpatients, with an average of 16.73. In the overall observation period and 12 months after the onset, the average frequency of outpatient services in the death group was more frequent than people in the survival group (33.87 vs. 27.90, 20.63 vs. 16.19, respectively).

Regarding hospitalization services, 994 (79.1%) cases received hospitalization services. The average frequency of hospitalization services was 4.08. Among them, the average frequency of hospitalization services for survival and death cases was 3.39 and 8.47, respectively. The ALOS of death cases was 15.61 days, whereas those who survived had an ALSO of 13.40 days. A total of 799 (71.7%) cases were hospitalized within 12 months after the onset, and the average frequency of hospitalization services for survival and death cases was2.73 and 4.62. The results of statistical tests show that utilization of medical services of the death cases was much higher than that of the survival cases, which are summarized in Table 2.

### 3.3. Survival Analysis of Cervical Cancer

In the study, cervical cancer cases were followed for 9–70 months, with a median follow-up of 35 months. The median survival was 67.47 months (95% CI = 66.88–68.06), with a three-year and five-year survival rate of 93.8% and 75.3%, respectively. The two-stage Cox proportional hazard model was performed, with death as the outcome variable. The univariate Cox proportional hazard model explored the association between age at onset (*p* < 0.05), ethnicity (*p* = 7.07), salary income in the year of onset (*p* = 0.417), the frequency of outpatient services (*p* = 0.410), the frequency of hospitalization services (*p* < 0.05), the ALOS (*p* < 0.05), metastasis (*p* < 0.05) and the death risk of cervical cancer. Then, the significant factors (*p* < 0.1) in univariate analysis were included in the multivariate Cox proportional hazard model. The results of multivariate analysis are summarized in Table 3. After potential confounders were controlled, the results showed that the age at onset (HR = 1.037, 95% CI = 1.024–1.051), the frequency of hospitalization services (HR = 1.085, 95% CI = 1.061–1.109), and the ALOS (HR = 1.020, 95% CI = 1.005–1.037) were the independent factors for death from cervical cancer.

### 3.4. Burden Analysis

Direct economic burden. The direct economic burden in the study was composed of outpatient and hospitalization expenses. The total outpatient cost of cervical cancer was $4.5 million, and the average was $4314, of which the average costs of survival and death cases were $4049 and $5063, respectively. Meanwhile, the total hospitalization cost of cervical cancer was $11.93 million, and the average was $12,007, of which the average hospitalization costs of survival and death cases were $9350 and $29,057, respectively. The average outpatient cost within 12 months after the onset was $2871 (survivors: $2746 and deaths: $3759) and hospitalization cost was $8963 (survivors: $7787 and deaths: $15,817). The average outpatient and hospitalization costs of death cases were higher than those of survivors. The results show that there are significant differences, which are summarized in Table 4.

Potential indirect burden. According to the “13th five year” population development plan issued by the city’s government, the life expectancy of residents in 2015 was 82.19 years; thus, the study selected 82.19 years as the life expectancy. The direct method was used to calculate the PYLL. After excluding the negative value, the total PYLL was 3605.51 years, and the average PYLL per case was 27.95 years. The potential economic loss was assessed by the salary income in the year of onset. Using the direct method, the total potential economic loss was $12.29 million, and the average potential economic loss per case was $95,200.

## 4. Discussion

These findings suggest that the epidemiological and economic burden of cervical cancer cannot be ignored. The age at onset, the frequency of hospitalization services and the ALOS were the related factors that increase the death risk of cervical cancer cases. In addition, the study found that the population had higher PYLL, direct economic burden and indirect economic loss caused by premature death.

The incidence rate and mortality of cervical cancer in the study were 14.85/100,000 and 1.82/100,000, respectively. This rate is slightly lower than the incidence rate and mortality (15.17/100,000 and 3.98/100,000, respectively) of Chinese female cervical cancer patients reported by Song et al. [18] in the same period. The review of the burden of cervical cancer in China [6] reported that the incidence rate and mortality in cities were 10.7/100,000 and 2.5/100,000, which showed an upward trend year by year. The study population had a stable source of income and a relatively better education, and could better obtain relevant health care knowledge, medical services and a cleaner living environment, which may have contributed to the discrepancy in the observed incidence rate and mortality [19,20,21]. As for the age of onset, similar to the studies in China [22], the incidence population was mainly concentrated in the 45–59 age group, and the tendency was younger. A previous study [23] indicated that the main peak of onset age has moved from the group aged 55–59 years to the group aged 45–49 years in rural China. In the study, the proportion in the 30–44 age group reached 36.4%. The younger age of onset would inevitably increase the burden. Therefore, in order to further improve the quality of life of young cervical cancer patients, more attention and resources should be invested in young women. For the utilization of outpatient and hospitalization services, both in the overall observation period and in the 12 months after the onset, the study found the utilization frequency of death cases, especially the ALOS and the hospitalization frequency, were higher than survival cases. These findings indicate that the death cases may have greater medical demands from diagnosis, which suggest that policy-making departments should take into account the surging medical demands of cervical cancer cases in the 12 months after onset and provide more comprehensive and accessible medical services.

The three-year survival and five-year survival rates of cervical cancer were 93.8% and 75.3%, respectively. The rates are consistent with the results of the study in Zhejiang Province, China [24], which reported that the five-year survival rate of cervical cancer in the city was 77.87%, but it was higher than the rate reported in a full population study in China from 2012–2015, which was 59.8% [25]. However, the study on global cancer survival analysis [26] showed that the five-year survival of cervical cancer in Iceland, Norway and South Korea reached more than 70% in 2005–2009, and a study in Korea [27] showed that the rate was 80–81.2%. The above results suggest that the five-year survival of cervical cancer in this study is slightly different, and the risk factors affecting survival need to be further analyzed. The study explored the factors affecting death from cervical cancer through the multivariate Cox proportional hazard model. The results showed that age at onset (HR = 1.037, 95% CI = 1.024–1.051), the frequency of hospitalization services (HR = 1.085, 95% CI = 1.061–1.109), and the ALOS (HR = 1.020, 95% CI = 1.005–1.037) were related to death from cervical cancer. Previous studies [28,29] are consistent with the study, showing that increasing age is an independent risk factor that predicts a poor overall survival trend. Government departments and hospitals should focus their attention on older patients in order to improve their living conditions. In addition, the study found that the cases with more hospitalizations and longer ALOS were related to death from cervical cancer. The reason may be that these cases have more serious conditions, poorer FIGO stage and more complications, which lead to having to accept more and longer hospitalization services [30,31,32,33]. Despite the increasing frequency of hospitalization services, the actual efficacy is not as good as expected, and the mortality risk is still greater than that of non-hospitalized patients. Therefore, when dealing with patients with a large number of hospitalizations and a long ALOS, more meticulous care should be given.

The average direct economic burden in the study was $14,981, which was basically in line with the direct economic burden of relatively developed regions in China reported in previous studies [34], such as Guangdong Province ($11,725) and Beijing ($18,795). However, a previous study in Eswatini [15] and the United States [35] reported that the cost burden of cervical cancer treatment was $12,707 and $10,031, respectively, which is lower than that reported in the study. In the study, the outpatient and hospitalization costs were $2871 and $8963 within 12 months after onset, which is lower than the average medical cost ($12,084) of cervical cancer patients reported in a previous study in the commercial insurance population of the United States [36]. A survey of the medical insurance population in Texas [37] found that the direct cost of patients in the first and second years after diagnosis of cervical cancer were as high as $50,846 and $27,656. Therefore, the costs in this study are lower in the shorter treatment time (12 months after the onset), and the costs of overall treatment are higher, which suggests that doctors can avoid greater costs by providing adequate treatment for patients within 12 months after onset. In addition, regarding the indirect burden, the average PYLL in the study reached 27.95 years, and the average potential economic loss reached $95,200, which was also far higher than that reported in Taiwan [38] and the United States [39]. In China, a previous study [40] on cervical cancer found that it was the cancer with the highest life loss, which also means that the onset age of cervical cancer may be getting younger and younger. All these suggest that we should increase investment in prevention activities, such as screening and early diagnosis. Government departments should provide more resources for patients at the early stages of cervical cancer, so as to prolong their survival period, improve their quality of life, and reduce demands caused by subsequent treatment.

The study data come from the authoritative data of the medical insurance department, which are clear, complete and accurate, with almost no missing observations. The data could show the real medical conditions of cases. Existing studies [41] have shown that good and high-quality medical insurance services can reduce the disease and economic burden of cervical cancer. More convenient and accurate medical insurance services should be provided for the wider population. Meanwhile, the epidemiological and economic burden of cervical cancer described in the study can guide the government to focus on the trend of younger onset age of cervical cancer and the treatment in the 12 months after onset, move resources towards the early diagnosis and treatment of cervical cancer, and formulate more accurate and practical policies. In addition, the study may show that it is realistic and cost-effective for the government of China to incorporate HPV vaccines into the national immunization program, which can improve the public’s awareness of cervical cancer and enable them to accept early screening and a healthier lifestyle [42,43,44].

The study is subject to several limitations. First, due to the lack of sufficient data, the study only assessed the direct economic burden data and the economic loss caused by death but did not consider non-medical expenses in the direct economic burden, resulting in an incomplete evaluation of the study. Second, due to the requirement of data confidentiality, the local government did not provide the patient’s out of pocket portion, where patients paid for medical treatment at their own expense, which may underestimate the actual economic burden. Third, as the burden of cervical cancer varies greatly among regions with different levels of development in China, the study was only aimed at the UEBMI population in a city of eastern China, which may lead to underrepresentation of the population and limit the extrapolation of data. Future study needs to further explore the burden of cervical cancer in a wider range from a more comprehensive perspective, and provide more accurate reference data for subsequent policy formulation.

## 5. Conclusions

This study was conducted to investigate epidemiological and economic burden of cervical cancer among the UEBMI population. The incidence rate of cervical cancer in this study was relatively high and the mortality rate was relatively low, which also could indicate that the five-year survival rate of cervical cancer patients found in the study was high. The age of onset was mainly distributed in two age groups, 30–44 (36.4%) and 45–59 (46.5%), with an obvious trend of younger age. For medical service utilization and expenses, the study found that death cases were higher than survival cases. The study also found that the age of onset, the frequency of hospitalization services and the ALOS were factors related with death from cervical cancer. In addition, PYLL and indirect economic burden caused by premature death in the study were at a high level. Policymakers, especially government officials, should increase publicity and capital investment to provide more convenient medical services and better reimbursement policies. Researchers should focus on the early abnormalities of younger women and further explore measures to reduce the incidence rate of cervical cancer. Doctors should make more accurate treatment choices for patients within the 12 months after onset and pay more attention to patients who experience greater frequency of hospitalization services and longer ALOS. Investors, especially HPV vaccine suppliers, should cooperate with the government to actively develop cervical cancer vaccines suitable for the Chinese population and ensure that individuals have sufficient opportunities to be vaccinated early.

## Figures and Tables

**Table 1 healthcare-11-00144-t001:** Demographic characteristics of cervical cancer cases.

Variable	Stage	Total (n)	Proportion (%)
Age of onset	<30	31	2.8
	30~44	518	36.4
	45~59	406	46.5
	60~74	131	11.7
	≥75	29	2.6
Ethnicity	Han	1094	98.1
	Non-Han	21	1.9
Income in the year of onset ($)	<5000	932	83.6
	5000~8000	82	7.4
	>8000	101	9.1
Metastasis	Yes	56	5.0
	No	1059	95.0

**Table 2 healthcare-11-00144-t002:** Medical service utilization of cervical cancer cases from 2010 to 2014.

	Variable	Total	Survivors	Deaths	*Z* Value	*p* Value
Outpatient service	Overall average frequency	28.63	27.90	33.87	2.92	0.003 *
	Average frequency within 12 months after onset	16.73	16.19	20.63	2.40	0.017 *
Hospitalization service	Overall average frequency	4.08	3.39	8.47	11.52	*p* < 0.001 *
	ALOS	13.70	13.40	15.61	4.21	*p* < 0.001 *
	Average frequency within 12 months after onset	3.01	2.73	4.62	6.88	*p* < 0.001 *

* Represents a statistically significant *p*-value.

**Table 3 healthcare-11-00144-t003:** Multivariate Cox proportional hazard model showing the variables associated with death from cervical cancer.

Variable	β	Wald	*p*	HR	95% CI
Age at onset	0.037	32.12	<0.001 *	1.037	1.024–1.051
Frequency of hospitalization services	0.082	52.89	<0.001 *	1.085	1.061–1.109
ALOS	0.020	6.430	0.011 *	1.020	1.005–1.037
Metastasis	0.392	2.264	0.132	1.479	0.888–2.464

* Represents a statistically significant *p*-value.

**Table 4 healthcare-11-00144-t004:** Analysis of direct economic burden of cervical cancer cases.

	Variable	Total	Survivors	Deaths	Z Value	*p* Value
Outpatient costs	Overall average cost	4314	4049	5063	4.38	*p* < 0.001 *
	Average cost within 12 months after onset	2871	2746	3759	3.68	*p* < 0.001 *
Hospitalization costs	Overall average cost	12,007	9350	29,057	12.07	*p* < 0.001 *
	Average cost within 12 months after onset	8963	7787	15,817	7.07	*p* < 0.001 *
Total		14,981	12,227	34,439	12.02	*p* < 0.001 *

* Represents a statistically significant *p*-value.

## Data Availability

Data available on request due to restrictions. The data presented in this study are available on request from the corresponding author. The data are not publicly available due to the restrictions of the medical insurance department.

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
