# Peer review of "Epidemiological/Disease and Economic Burdens of Cervical Cancer in 2010–2014: Are Younger Women at Risk?"

_healthcare, 2023, doi:10.3390/healthcare11010144_

Round 1
Reviewer 1 Report
Very well written and is an interesting paper. Here are my comments.
Since this paper does not have a conjecture or have a development of hypothesis in the text. I therefore wonder the position of this paper in the literature.
I do not see what the conjecture that the authors propose on the relation between cancer risk and age, also I cannot see how the conjecture is developed.
Since the development of hypothesis is important to convince the readers about story, I would suggest to further expand the development of hypothesis.
Also, the conclusion section may further be enriched by adding/discussing the implications of findings to policy makers, researchers, and investors.
Digits should be consistent in each tables.
In some tables, we don't know what is your aim. Results should be discussed rather than being reported.
Can you test the statistical difference between age groups in Table 1?
Are there differences within between outpatient service groups in Table 2?
What is the definition of "overall average cost" in Table 4?
Reviewer 2 Report
Dear authors,
Congratulations on your work. You have put much effort into recruiting a significant sample size. However, I have some comments and questions for you to approach, discuss and include in the text to improve your paper. Your readers will be able to get more information about your research for potential replication or application in clinical settings.
Methods:
- What design was followed for the research?. What was the process of the study throughout the study (2010-2014)?. Perhaps it would be useful to add a flow chart that also visually explains the process.
- It is not clear to me what kind of ethical protocol has been followed in the study. Data were collected from the system, but did the participants sign an informed consent form to agree to participate in the study? Is the study registered? Is it approved by a bioethics committee?.
- It is also not specified whether the study follows standards and guidelines, such as CONSORT (trial reporting).
- In general, I believe that the methodology needs to be improved.
Conclusions:
- The reflections in the conclusions section do not seem to me to provide new knowledge gained from the facts of the experiment.
Round 2
Reviewer 2 Report
Dear authors,
The improvement made to the manuscript is substantial. Therefore, my recommendation would be to publish the article in its current version.
Congratulations on the work done!
Kind regards